# Using a Linear Probe Ultrasound for the Detection of First-Trimester Pregnancies in the Emergency Department

**DOI:** 10.3390/diagnostics13152564

**Published:** 2023-08-01

**Authors:** Soheil Saadat, Michelle Thao Nguyen, Isabelle Nepomuceno, Erinna Thai, Ami Kurzweil, Heesun Choi, Shadi Lahham, John Christian Fox

**Affiliations:** 1Department of Emergency Medicine, School of Medicine, University of California, Irvine, CA 92697, USA; inepomuc@hs.uci.edu (I.N.); jfox@hs.uci.edu (J.C.F.); 2School of Medicine, University of California, Irvine, CA 92697, USA; mtnguy10@hs.uci.edu; 3College of Osteopathic Medicine of the Pacific, Western University, Pomona, CA 91766, USA; erinna.thai@westernu.edu; 4Department of Emergency Medicine, Eisenhower Health, Rancho Mirage, CA 92270, USA; akurzweil@eisenhowerhealth.org; 5Department of Emergency Medicine, Kingman Regional Medical Center, Kingman, AZ 86409, USA; heesunchoido@gmail.com; 6Department of Emergency Medicine, Kaiser Permanente, Irvine, CA 92618, USA; slahham8@gmail.com

**Keywords:** ultrasound, first-trimester intrauterine pregnancy, linear probe ultrasound, transabdominal ultrasound, transvaginal ultrasound

## Abstract

Linear probe point-of-care ultrasound (LPUS) presents a less invasive alternative for identifying intrauterine pregnancies (IUPs) compared to usual practice (transabdominal (TAUS) or transvaginal (TVUS) ultrasound). TAUS and TVUS can be invasive or produce lower-resolution images than LPUS. The purpose of this study is to determine whether a linear probe alone can identify first-trimester IUPs. A convenience sample of 21 patients were enrolled at the University of California Irvine ED during a 7-month period. The inclusion criteria were English- or Spanish-speaking women (≥18 years) in their first trimester of pregnancy (≤12 weeks pregnant) with a body mass index (BMI) of <35. The exclusion criteria were psychiatric, incarcerated, or cognitively impaired patients. An ED physician performed LPUS and ordered a confirmatory ultrasound. The 21 patients enrolled had a mean age of 28.6 ± 6.60 years, BMI of 26.6 ± 5.03, and gestational age of 7.4 ± 2.69 weeks. Considering the 95% confidence interval, we are 97.5% confident that the sensitivity and specificity of LPUS to identify IUPs does not exceed 67.1% and 93.2%, respectively. Our pilot data did not demonstrate that LPUS can independently visualize IUPs in first-trimester patients.

## 1. Introduction

Approximately 500,000 women per year in the United States come to the emergency department (ED) with complaints of vaginal bleeding or pelvic pain, making pregnancy-related problems the fifth most common reason for ED visits [1,2]. Other common pregnancy-related complaints among pregnant women visiting the ED include abdominal pain, fever, and vomiting [3]. In total, 50% of these women presenting with vaginal bleeding are at high risk of miscarriage. Due to the prevalence of women entering the ED with these problems, it is imperative to quickly diagnose these patients in order to prevent further complications like ectopic pregnancy, ovarian torsion, threatened abortion, or tubo-ovarian abscesses [4]. As such, ED physicians must diagnose promptly and follow-up with patients to prevent further complications, including preterm delivery and low birth weight [1].

The usual practice for diagnosing first-trimester pregnancies is the measurement of human chorionic gonadotropin (hCG) serum concentration. HCG values can vary widely among individuals; studies have suggested that serum hCG levels below 5 mIU/mL are associated with nonviable pregnancies, while hCG levels above 25 mIU/mL indicate potentially viable pregnancies [5]. When hCG levels reach 1500 mIU/L, the vast majority of intrauterine pregnancies (IUPs) can be visualized and localized using ultrasonography [6]. Combining both hCG measurements and pelvic ultrasounds provides a more comprehensive evaluation of a pregnancy and guides treatment.

Pelvic ultrasounds, commonly used for visualizing pregnancy, can be performed using two main techniques: transabdominal ultrasound (TAUS) and transvaginal ultrasound (TVUS). During a TAUS, a curvilinear probe is placed on the skin in the suprapubic region [5]. The curvilinear probe utilizes a frequency of 2–5 MHz, which provides a large field of view of the pelvis with great penetration. One disadvantage is that it provides relatively lower resolution, resulting in less detailed images. TVUS, on the other hand, involves inserting an endocavitary probe into the vaginal vault. It utilizes higher frequencies in the 5–7.5 MHz range and thus has superior resolution but a narrower field of view [7,8]. Although TVUS is the preferred method for pregnancy screening, it is worth noting that it is more invasive and more time-consuming in the ED than TAUS. Previous studies have demonstrated that TAUS and TVUS have shown high sensitivity and specificity. In one study comparing TAUS and TVUS, TAUS had a 91% sensitivity and 83% specificity, while TVUS had a 96% sensitivity and 89% specificity [8].

While TVUS and TAUS can effectively identify first-trimester IUPs, linear probe point-of-care ultrasound (LPUS) presents a quicker and less invasive alternative [9]. Linear probes have a frequency of 4–13 MHz, which surpasses the frequencies of both curvilinear and endovaginal probes. At the expense of penetration, the linear probe visualizes superficial layers of the skin to produce higher-resolution images [7]. A larger fat percentage remains a concern when using LPUS in detecting IUPs because a greater body mass index (BMI) can decrease ultrasound wave penetration and minimize visibility.

Considering the prevalence of obesity, it is crucial to assess the feasibility of LPUS in this population. At the conception of this study, ~1/5 and ~1/10 women in the United States of America have a BMI of >30 and >35, respectively [10]. In a retrospective study conducted in 2013, physicians consistently rated ultrasound images of overweight patients with the highest quality, while images of obese patients had the lowest quality. Patients with an average BMI fell somewhere in between [11]. Further investigation is needed to determine if there is an optimal BMI range for utilizing LPUS.

Recent studies have found that first-trimester IUPs can be identified using TAUS followed by LPUS [12]. The objective of this pilot study is to evaluate whether LPUS alone is a viable tool to identify IUPs for adult patients presenting to the ED with a first-trimester pregnancy.

## 2. Materials and Methods

### 2.1. Study Design and Setting

We performed a prospective, observational single-site pilot study on a convenience sample of patients who presented to the ED between August 2019 and May 2020. Data were collected at our urban level 1 trauma center with an annual volume of 65,000 patients. All experiments were carried out according to the Institutional Review Board of the University of California, Irvine, and were consistent with federal guidelines. All data collection was conducted at the Emergency Department of the University of California Irvine Medical Center in Orange, CA.

### 2.2. Selection of Participants

Research associates of the Emergency Medicine Research Associates Program screened the ED trackboard to enroll eligible patients in this study. Research associates were present Monday through Sunday from 8:00 am to 12:00 midnight. Patients were eligible for inclusion if they were English- or Spanish-speaking, ≥18 years of age, presented to the ED with suspicion of a first-trimester pregnancy (≤12 weeks), and had a BMI < 35. The exclusion criteria were cognitively impaired, incarcerated, or psychiatric patients and those diagnosed with ectopic pregnancy. Patients were screened for pregnancy-related complaints including vaginal bleeding or abdominal pain. If a patient met all the criteria, they were then approached and informed of what their role would be in the study: to receive both a transabdominal ultrasound with a linear transducer and a confirmatory ultrasound, which is the usual practice for visualizing IUPs. The usual practice included TVUS or TAUS. Once the patient agreed to participate, they consented by signing a Health Insurance Portability and Accountability Act (HIPAA) form and consent form. The patient was also given a copy of her forms.

### 2.3. Study Protocol

ED physicians or research associates approached patients in their first trimester of pregnancy for enrollment in the study. Screening criteria included that the patient was ≤12 weeks pregnant with a body mass index (BMI) <35. Patients were determined to be ≤12 weeks pregnant based on the date of their last menstrual period. Following verbal and written consent, researchers collected data using a systematic approach on a standard data abstraction sheet. Collected data included age, height, weight, BMI, blood hCG concentration, and date of LMP.

ED physicians exclusively performed LPUS using a Mindray TE-7 machine in the abdominal software setting before ordering the usual practice ultrasound. The department of radiology conducted the usual practice ultrasounds, which consisted of either transvaginal ultrasound using an endovaginal probe or transabdominal ultrasound using a curvilinear probe. The choice of probe was determined by the ultrasound technician.

It is important to note that all ED physicians are credentialed by the hospital for point-of-care ultrasound (POCUS) of the female pelvis during the first trimester of pregnancy. This credentialing process ensures their competence in performing these ultrasounds. Emergency medicine residents, fellows, and attendings undergo comprehensive training and gain significant experience in all types of POCUS, with accreditation from the Residency Review Committee for Emergency Medicine and the Emergency Ultrasound Fellowship Accreditation Council, guaranteeing adherence to rigorous standards. Furthermore, all physicians in the ED are board-certified and possess extensive ultrasound experience.

The primary outcome of interest in this study was the presence of an intrauterine pregnancy (IUP), and additional collected data included gestational age and the type of ultrasound performed. The physician performing the ultrasound recorded all the collected data. To ensure unbiased interpretation of the ultrasounds, if the usual practice ultrasound was performed first, the physician remained blinded to the results until they conducted the transabdominal ultrasound with the linear transducer. Following the completion of the ultrasounds and data collection, the physician received a USD 5 gift card as compensation for their time and contribution to the study.

### 2.4. Statistical Analysis

Sensitivity, specificity, positive predictive values, and negative predictive values are presented as percentages and 95% confidence intervals (CIs). STATA 14 (StataCorp. 2015. Stat Statistical Software: Release 14. College Station, TX, USA: StataCorp LP) was used for data analysis and the “diagti” command was utilized to calculate test specifications and confidence intervals.

## 3. Results

Initially, 28 patients were approached for enrollment in this study; however, 7 patients declined to participate. A total of 21 patients were enrolled in the study, 13 (61.9%) of whom had the chief complaint of vaginal bleeding or abdominal pain. Other chief complaints included motor vehicle collision, leg pain, ectopic pregnancy, dizziness, nausea, vomiting, and pelvic pain. The enrolled patients had a mean age of 28.6 ± 6.60 years old, BMI was 26.6 ± 5.03, and gestational age was 7.4 ± 2.69 weeks. An IUP was confirmed in 16 cases using ultrasound, not detected in 4 cases, and inconclusive in 1 case (Figure 1).

LPUS performed by ED physicians had a sensitivity of 41.2% (18.4–67.1%) and a specificity of 50.0% (6.8–93.2%) in detecting an IUP. By considering the “inconclusive IUP” as a negative case, the sensitivity was 43.8% (19.8–70.1%) and the specificity was 60.0% (14.7–94.7%) (Table 1).

## 4. Discussion

This study aimed to determine if LPUS can independently detect first-trimester IUPs. Our pilot data did not support that LPUS alone can reliably identify IUPs during the first trimester.

Our pilot data demonstrate that out of 21 patients, 16 IUPs were confirmed through the usual practice using ultrasound. In contrast, only 9 IUPs were identified using LPUS. Despite the higher resolution of the linear probe, it is unclear why fewer IUPs were identified compared to the usual practice using ultrasound. Several factors could have contributed to the discrepancy between the two modalities, including the level of skill and training of the provider in identifying IUPs with a LPUS and anatomical variations in the patient population.

The single case with an inconclusive result could be interpreted as indicative of either a positive or negative pregnancy. To prioritize patient safety and avoid a potential missed diagnosis, Case 1 of Table 1 may better represent the diagnostic value of LPUS in the first trimester. The 95% confidence interval for LPUS sensitivity was 18.4–67.1%. Thus, the possibility for the true sensitivity to exceed 67.1% is only 2.5% [(100–95%)/2]. Even in large studies, it is unlikely that the sensitivity of LPUS exceeds 67.1%, which is not acceptable for a diagnostic test in clinical settings.

Similarly, we can be 97.5% confident that the true specificity of LPUS in detecting IUPs in the first trimester is less than 93.2%. Compared to TAUS (sensitivity of 91%, specificity of 83%) and TVUS (sensitivity of 96%, specificity of 89%), LPUS alone does not seem clinically applicable in visualizing IUPs. The gestational age is a crucial factor in the detectability of IUPs using ultrasound. Our study included all first-trimester IUPs with a mean gestational age of 7.4 ± 2.69 weeks. However, we did not evaluate the accuracy of LPUS based on the exact gestational age. While LPUS may not be the best diagnostic tool for early IUPs, it could still be clinically useful in ruling out IUPs in the later weeks of pregnancy, even beyond the first trimester. Further studies are needed to determine the accuracy of LPUS in weeks 8 and later.

The advantage of using LPUS in addition to another modality is that it can penetrate superficial layers of the skin to produce a higher-resolution image due to its higher frequency (4–13 MHz) compared to the curvilinear probe (2–15 MHz). The limitation of this probe is that it cannot penetrate through deep layers of fat [7]. Previous studies have identified that the linear probe can reach a depth up to 4.1 cm [12]. Another study concluded that LPUS could visualize IUPs in women with anteverted uteri at depths ≤ 6 cm [13]. There is no general agreement on the threshold at which LPUS can identify IUPs; thus, further studies are merited to determine this threshold.

The visibility of an IUP using LPUS is dependent on a myriad of factors, including, but not limited to, the depth of the uterus from the skin and the patient’s body composition, as a thick layer of skin or adipose tissue can create a barrier for ultrasound penetration [12]. Additionally, at early stages, the fetus may be difficult to distinguish from surrounding tissue [12]. Obtaining an image with a linear probe is difficult for a retroverted uterus and deeper organs. Considering these factors, LPUS alone may not be the best diagnostic tool for early IUPs.

The data were collected from a single site, and the findings from this site may not be generalizable to other patient populations and could have introduced selection bias due to convenience sampling. Additionally, operator experience may affect measurements as our study did not seek to determine the amount of physician training required to identify an IUP using LPUS. Interrater reliability was not measured in this study. The distribution of BMI and gestational age was not uniform in our sample; therefore, we cannot study the accuracy of LPUS according to BMI or gestational age.

Previous studies support the utilization of LPUS to visualize an IUP following a failed curvilinear ultrasound [12]. Though LPUS has been shown to potentially improve IUP diagnosis, it is unclear if LPUS can independently and reliably detect IUPs [14].

Future studies are needed as our pilot data did not demonstrate the feasibility of using LPUS alone to identify IUPs. To further explore the potential of LPUS for detecting IUPs in first-trimester pregnancy with respect to BMI, future studies should consider patients with a larger range of BMIs across several locations. This would provide a more diverse patient population and a wider range of operator experience. Considering the significant and growing proportion of obese women, it is crucial for future studies to investigate whether LPUS remains a viable option for pregnant patients with BMIs exceeding 35. Moreover, subsequent research endeavors could explore the interrater reliability of LPUS and assess how operator experience influences its diagnostic accuracy.

## 5. Conclusions

Our data did not demonstrate that LPUS can independently be used to visualize an IUP in first-trimester patients. Future studies are needed to determine the discriminatory threshold of BMI and gestational age at which LPUS can be used.

## Figures and Tables

**Figure 1 diagnostics-13-02564-f001:**
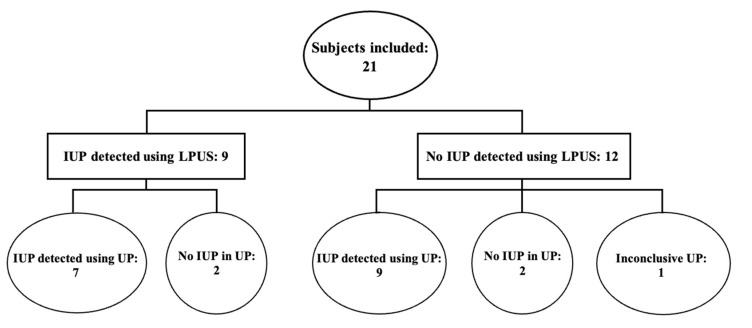
Breakdown of intrauterine pregnancy (IUP) detected using linear probe ultrasound (LPUS) and the usual practice (UP).

**Table 1 diagnostics-13-02564-t001:** Test specifications of LPUS for detecting IUPs *.

Test Specification	Case 1: If the Inconclusive Case is Considered as Pregnant	Case 2: If the Inconclusive Case is Considered as Non-Pregnant
Sensitivity	41.2% (18.4–67.1%)	43.8% (19.8–70.1%)
Specificity	50.0% (6.8–93.2%)	60.0% (14.7–94.7%)
Positive predictive value	77.8% (40.0–97.2%)	77.8% (40.0–97.2%)
Negative predictive value	16.7% (2.0–48.4%)	25.0% (5.5–57.2%)

* Table entries are mean (95% confidence intervals).

## Data Availability

The data presented in this study are available on request from the corresponding author. The data are not publicly available due to protected patient information.

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
