# Peer review of "Using a Linear Probe Ultrasound for the Detection of First-Trimester Pregnancies in the Emergency Department"

_diagnostics, 2023, doi:10.3390/diagnostics13152564_

Round 1

Reviewer 1 Report

This is an interesting study that warrants publication. I have a few minor remarks.

Please provide a Table describing the abbreviations.

Please check the reporting of the references.

Reviewer 2 Report

The data of this brief report come from a single site with a limited number of patients. The training of emergency physicians was not controlled and interrater reliability was not assessed, as mentioned in the discussion. However, ultrasound techniques essentially rely on proper training. Whether the physician was familiar to LPUS is not reported in the manuscript. Consequently, the parameters of this study are poorly controlled. The reviewer considers that, at a minimum, the completion of adequate training must be demonstrated and inter-observer variability should be evaluated, e.g. 1-1 agreement with the senior gynecologist on 3 selected cases.

Please specify in the introduction that, at the time of conception, ~1/5 and ~1/10 of women in the USA have a BMI >30 and >35 respectively. Prospects for adapting technologies to overweight patients need to be discussed, apart from adjusting the depth threshold in fat layers.

Please provide an illustration for LPUS, ideally in relation to an illustration for TAUS or TVUS.

Reference 6 is a bit outdated. Improper use of reference 10 in the introduction.

Correct use of the English language. Minor inconsistencies in layout (spacing).

Author Response

Please see the attachment. Minor inconsistencies in the layout has been corrected.

Reviewer 3 Report

In this paper, the authors present the supremacy of using a linear probe ultrasound for the detection of first trimester pregnancies in the emergency department, compared to the abdominal and transvaginal ultrasound. The subject is of great interest and can lead to a quicker and more reliable diagnosis of intracavitary pregnancies.
The introduction focuses on the general knowledge on the subject. The information provided is useful in understanding the topic and the importance of the subject in nowadays medicine.
The methods used are complex and reliable.
The inclusion and exclusion criteria were wisely chosen.
The authors conducted a proper collection of the data. The information presented is up to date, suitable, and substantial.
The conclusions are coherent and sustain the findings.
The figures and tables presented are easy to interpret and understand.
Good English level.
I recommend it for publication.

Author Response

Thank you for your response. Based on the comments and suggestions provided by Reviewer 3, it appears that no revisions need to be addressed for the manuscript.

Round 2

Reviewer 2 Report

The authors have addressed the criticisms considering the nature of this brief report.

For the illustration of methods, an original figure depicting the quality of image obtained during the study would had been more relevant.

Author Response

We are unable to provide an original figure illustrating the image quality. However, we would like to emphasize that all images included in the study have undergone thorough review by a board-certified emergency physician. We have taken diligent measures to address all other concerns raised during the evaluation process. Thank you for your understanding and consideration.